# Impact of youth lay health workers on HIV service delivery in South Africa: A pragmatic cluster randomized trial of the Youth Health Africa program

**Deanna Tollefson**[1,2]*, **Sayan Dasgupta**[2], **Geoffrey Setswe**[3,4], **Sarah Reeves**[5], **Salome Charalambous**[3,6], **Ann Duerr**[1,2]

**1** Department of Global Health, University of Washington, Seattle, Washington, United States of America, **2** Fred Hutchinson Cancer Research Center, Vaccine and Infectious Disease & Public Health Science Divisions, Seattle, Washington, United States of America, **3** The Aurum Institute, Implementation Research Division, Parktown, Gauteng, South Africa, **4** Department of Health Studies, University of South Africa, Pretoria, Gauteng, South Africa, **5** Youth Health Africa, Parktown, Gauteng, South Africa, **6** School of Public Health, University of the Witwatersrand, Johannesburg, Gauteng, South Africa

* dtollefs@fredhutch.org

**Data Availability Statement:** All relevant data are within the paper and Supporting information files.

## Abstract

### Background

Innovative approaches are needed to increase lay health workers in HIV programs. The Youth Health Africa (YHA) program is a novel approach that places young adults seeking work experience in one-year internships in health facilities to support HIV-related programming (e.g., HIV testing) or administration (e.g., filing).

### Methods

We implemented a pragmatic, randomized trial among 20 facilities in Ngaka Modiri Molema district in North West province from October 2020-August 2021 to assess impact of YHA interns on HIV testing, treatment initiation, and retention in care. The primary outcome was proportion of patients tested for HIV. Secondary outcomes assessed HIV positivity, initiation in care, retention in care, and HIV testing among males and adolescents/young adults. We conducted an intention-to-treat analysis accounting for variations in baseline outcomes between control and intervention facilities using difference-in-difference and controlled time series approaches. We repeated this using as-treated groupings for sensitivity analyses.

### Results

Fifty interns were placed in 20 facilities; thirty-four interns remained at 18 facilities through August 2021. Compared to control facilities, intervention facilities had a greater improvement in HIV testing (ΔΔ+5.7%, 95% Confidence Interval (CI): -3.7%-15.1%) and treatment initiation (ΔΔ+10.3%, 95% CI: -27.8–48.5%), but these differences were not statistically significant. There was an immediate increase in HIV testing in intervention facilities after program interns were placed, which was not observed in control facilities; this difference was

**Funding:** The authors received no specific funding for this work. This study was conducted on a routinely implemented program. Aurum Institute supported miscellaneous research costs not covered by routine program implementation. Fred Hutch paid for registration of the trial.

**Competing interests:** The Aurum Institute was a funder to the Youth Health Africa organization, and SR supported monitoring and evaluation activities for Youth Health Africa. This does not alter our adherence to PLOS ONE policies on sharing data and materials. No other authors have any conflicts of interest to report.

significant ($\Delta\Delta$+8.4%, 95% CI: 0.5–16.4%, p = 0.036). There were no other differences in outcomes observed between intervention and control facilities.

## Conclusion

This was largely a null trial, but there were signals that program interns may have positive impact on HIV testing and treatment initiation. As implemented in this study, addition of YHA program interns had little impact on facility-based HIV service delivery. A higher number of interns placed per facility may be necessary to affect HIV services.

## Trial registration

Registration: This trial was registered with the ISRCTN (Registration number: ISRCTN67031403) in October 2022.

## 1. Introduction

Lay health workers (LHWs), defined as persons who support health work without formal qualifications [1], offer great potential to promote health and reduce human resource challenges in the healthcare sector, especially in low and middle-income countries [2–5]. LHW programs are considered a critical tool to help HIV programs improve reach and service delivery [6–9], but innovative approaches are needed to expand the reach of LHW programs, which are often limited by political and financial constraints [10,11].

Youth Health Africa (YHA) is an innovative LHW approach that has showed potential to support HIV service delivery at health facilities in South Africa [12,13]. YHA places young adults as temporary LHWs at health facilities in communities implementing the program to support non-clinical tasks, ranging from HIV testing and counseling to filing patient records [12]. This program stands in contrast to traditional LHW programs in South Africa, as it operates as an internship program, with a primary goal to provide young adults workplace training to improve their future employability and reduce youth unemployment [11,14]. It is also primarily funded by private businesses in response to the country's Broad-based Black Economic Empowerment policy, whereas traditional LHW programs are government funded [11,15].

Rigorous evaluation is needed to assess the impact of YHA on health services to determine whether this LHW approach is effective in improving health care delivery, specifically in the HIV care sector. Past research has evaluated the YHA approach through observational studies [12,13], but the limitations to these methods are well established. We therefore conducted a pragmatic, cluster randomized trial to assess the impact that placement of youth interns in health facilities had on HIV service delivery.

## 2. Methods

### 2.1. Study design

We conducted a randomized trial in 20 facilities slated to begin the YHA program. All facilities received a minimum allocation of YHA interns; allocation of additional interns was timed so that half of the clinics also received program interns (see study arms) to allow assessment of the impact of YHA program interns on HIV testing, treatment initiation, and retention in care. This was a pragmatic trial as YHA implemented and monitored the program following its standard operations, and we measured impact using routinely reported facility-level HIV

data. The intervention was in place from October 2020–August 2021. The YHA program has been described in detail elsewhere [12,13].

## 2.2. Setting

This study was conducted in health facilities in Ngaka Modiri Molema (NMM) district, North West province, South Africa. NMM is a semi-urban district in a rural province with high youth unemployment [16]. NMM has been a focus district of the U.S. President's Emergency Fund for AIDS (PEFPAR) [17] and has a relatively high proportion of people who had been previously tested for HIV [18].

## 2.3. Study arms

This was a two-arm study, with 1:1 randomization at 20 purposively sampled health facilities. There were 10 facilities in each arm. All 20 facilities received a minimum package of 1–2 interns assigned to administrative roles, such as filing or data capture (henceforth called "admin interns"). This helped to ensure consistent data quality between intervention and control facilities; this was necessary to ensure changes we observed were due to an impact on HIV service delivery and not an artefact of improvements in data quality. Intervention facilities each received the minimum package plus the intervention package: 1–2 interns assigned to support programmatic roles, like HIV testing and counseling, patient navigating, and tracing (henceforth called "program interns"). The exact number and roles of interns were determined by each facility, based on their needs and size.

## 2.4. Site Selection and randomization

Health facilities were eligible for this study if they were in NMM district, had never received interns from YHA, collaborated with Aurum Institute as a PEPFAR implementing partner, and had a need for three interns (the minimum number of interns that would be placed in a clinic assigned to the intervention group). Twenty eligible facilities were identified and agreed to participate in this study. These facilities were randomized to the control or intervention group at a public randomization event in September 2020 attended by the research team, YHA program staff, facility leaders, and the NMM health district manager. There was no blinding in this study.

## 2.5. Outcomes

The primary outcome in this study was the proportion of people attending the clinic who were tested for HIV (% tested), but we also examined five other HIV service outcomes related to HIV testing, treatment initiation, and retention in care (Table 1). To further assess testing, we reviewed the proportion of those tested who were positive for HIV (% positive). To assess timely linkage to care, we reviewed the proportion of people testing positive who were initiated on treatment within 14 days (% initiated). To assess retention in care, we reviewed the proportion of people on treatment who did not return within 28 days of their scheduled appointment (% early default), the proportion of those on treatment who did not return within 90 days of their scheduled appointment (% late default), and the proportion of those on treatment that had been missing for more than 90 days since their last scheduled appointment (% loss to follow-up).

In addition, we explored whether the YHA program increased testing among young people and males, two groups that are traditionally under-tested. To do this, we examined the proportion of people tested for HIV who were adolescents or young adults (ages 10–29 years old), the

**Table 1. HIV service outcomes and their definitions.**

| Outcome | Calculation | Numerator | Denominator |
|---|---|---|---|
| *Testing* | | | |
| **% Tested**[*] | HTS_TST / Headcount | Number tested for HIV | Total headcount |
| **% Positive** | HTS_POS/ HTS_TST | Number testing positive for HIV | Number tested for HIV |
| *Treatment* | | | |
| **% Initiated in 14 days** | INITIATED_14DAYS / HTS_POS | Number initiating treatment within 14 days of diagnosis | Number testing positive for HIV |
| *Retention* | | | |
| **% Early Default** | ART_DEFAULT_EARLY / TX_CURR90 | Number who did not return for treatment within 28 days of appointment | Number on treatment |
| **% Late Default** | ART_DEFAULT_LATE / TX_CURR90 | Number who did not return for treatment within 89 days of appointment | Number on treatment |
| **% Loss to Follow-Up** | ART_DEFAULT_ULTF / TX_CURR90 | Number of patients out of care for ≥90 days with no outcome (i.e., on the unconfirmed lost to follow-up list) | Number on treatment |

[*]Primary outcome for which the study was powered.

proportion of people tested for HIV who were male, and the proportion of people tested for HIV who were male adolescents or young adults.

## 2.6. Data source

Outcome data came from facility-level data that were reported routinely to Aurum Institute from the facility through TIER.Net, South Africa's national HIV surveillance system. Headcount data were not available for one intervention facility from March-August 2021; missing values were imputed with the facility's monthly average headcount from the previous six months. Three facilities (two control, one intervention) were missing headcount data for November-December 2019; these values were imputed with each facility's monthly average headcount from the remaining pre-COVID period (October 2019-March 2020).

## 2.7. Data analysis

Prior to analysis, we assessed the similarity of facilities in the intervention and control groups at baseline (January–August 2020), comparing facility size, counts of patients served, and proportion of people tested for HIV, positive for HIV, and initiated on treatment.

We conducted two sets of analyses using intention-to-treat assignments. The first assessed the cumulative impact of the intervention through a difference-in-difference analysis; this was our primary analysis and was used to examine all outcomes described above. The second assessed impact by examining monthly variation through a controlled, interrupted time series analysis; this was our secondary analysis and was used to further examine HIV service outcomes only. We conducted all analyses in R 3.6.1 (R Core Team 2019, Vienna, Austria).

**Analysis 1.** We conducted difference-in-difference analyses using linear models to measure the difference in outcomes for the intervention group as compared to the control group from a baseline period (January-August 2020) to study period (January-August 2021). For these analyses, we designated the first three months of this intervention (October-December 2020) as a "run-in" period, hypothesizing it would take interns time to adjust to new roles and impact change; we excluded the run-in period from the analysis. We selected the baseline period to mirror the study period in calendar time. Testing and treatment outcomes were

aggregated across the eight-month baseline and study periods (e.g., % tested for HIV = total tested for HIV over 8 months / total headcount for 8 months). The denominator used to calculate retention outcomes could not be aggregated by month, so monthly means were calculated for the default and loss to follow-up outcomes for baseline and study periods.

Models for each outcome were as follows, with 'Time' representing the period (baseline or study), 'Treat' representing the treatment group (control or intervention), and 'DID' calculated as Time*Treat:

$$Y = \beta_0 + \beta_1 \text{Time} + \beta_2 \text{Treat} + \beta_3 \text{DID}$$

We were interested in $\beta_3$, which measured the difference in change between the baseline and study period for the intervention versus the control group (difference-in-difference). We tested the null hypothesis that there was no difference in change between the intervention and the control groups ($\beta_3 = 0$), which we assessed at the 0.05 significance level. We ran these models using the "stats" package in R.

**Analysis 2.**  We conducted controlled, interrupted time series analyses for HIV service outcomes using segmented linear regression models to assess differences in monthly proportions and trends between the intervention and control groups over time. For these models, we used data from a longer time period than the difference-in-difference analysis to establish trends; we included one-year pre-intervention (October 2019-September 2020) and the full intervention period (October 2020-August 2021). Time was measured in calendar months.

Models for each outcome were as follows, with 'Time' representing the number of months since the start of the baseline period, 'Intervention' representing the period (pre-intervention or intervention), 'TimeAfterIntervention' representing the number of months since the placement of the first interns (i.e., months since October 2020), and 'Treat' representing the treatment group (control or intervention).

$$Y_{ijt} = \beta_0 + \beta_1 \text{Time}_{ijt} + \beta_2 \text{Intervention}_{ijt} + \beta_3 \text{TimeAfterIntervention}_{ijt}$$
$$+ \beta_4 \text{Treat}_{ijt} + \beta_5 \text{Treat} * \text{Time}_{ijt} + \beta_6 \text{Treat} * \text{Intervention}_{ijt}$$
$$+ \beta_7 \text{Treat} * \text{TimeAfterIntervention}_{ijt} + E_{ij}$$

We were interested in the magnitude and significance of $\beta_6$, which measured the difference-in-difference for immediate change (i.e., the immediate difference observed after intern placement in the intervention group minus the immediate difference observed in the control groups) and $\beta_7$, which measured the difference-in-difference for slope change (i.e., the change in slope after the intervention was implemented for the intervention group minus the change in slope observed in the control group). For each outcome, we tested the null hypotheses that there was no difference in immediate change between groups ($\beta_6 = 0$) or difference in trends between groups ($\beta_7 = 0$), which we assessed at the 0.05 significance level. We ran these models using the "nlme" package in R [19], using restricted maximum likelihood estimates (REML) and adjusting for autocorrelation (AR1) and clustering at the facility level. We fit all models with random slopes and intercepts.

## 2.8. Sensitivity analysis

In a sensitivity analysis, we conducted the above-mentioned analyses using as-treated assignments, comparing control facilities to 'high' intervention facilities and 'low' intervention facilities. In the as-treated analysis, facilities were designated as controls if they never had program interns and had admin interns for at least three-quarters of the study period (six months). Facilities were classified as 'high' intervention if they had both admin and program interns

simultaneously placed for at least three-quarters of the study period. They were classified as a 'low' intervention if they had admin and program interns simultaneously placed, but for less than three-quarters of the study period. Facilities were excluded from the analysis if they did not meet one of these definitions. In a second sensitivity analysis, we re-ran the intention-to-treat and as-treated difference-in-difference and time series analyses for the outcome "% tested," excluding the intervention facility that required imputed headcount data for a portion of the intervention period.

### 2.9. Study power

This study had 80% power to detect a minimum of 33% change for HIV testing (the primary outcome) if the intraclass correlation was 0.01 (i.e., change from 18.4% people tested to 24.5% of people tested), or a minimum of 82% change if the intraclass correlation was 0.05 (i.e., change from 18.4% of people tested to 33.4% of people tested); these calculations assumed a 5% level of significance. Power was calculated using baseline HIV testing data for all included clinics.

### 2.10. Ethics

This study was approved by the University of Witwatersrand (Johannesburg, South Africa) human research institutional review board (IRB) and the provincial ethics committee and registered with the ISRCTN (ISRCTN67031403). The IRB determined no written consent was necessary as health facilities (not individuals) were the subject and unit of analysis and the study required no study-specific data collection. Eligible facilities were invited to participate in the intervention and could decline without consequence. Interns participated in their work, under normal program circumstances; no study-specific data were collected on interns, negating their need for consent. This pragmatic trial was registered retrospectively, as it was instituted using an ongoing program and required no grant funding or prerequisites for registration. To the best of our knowledge, no additional trials are being conducted on the YHA program. Facilities in the control group were eligible to receive programmatic interns after completion of this study.

## 3. Results

### 3.1. Baseline characteristics

There were substantial differences among facilities in the control and intervention groups at baseline (Table 2). The intervention group included larger facilities than the control group and had higher mean counts and proportions for key indicators, but these differences were not statistically significant. Similar differences persisted at baseline in the as-treated analysis.

### 3.2. Intervention adherence

Fifty interns were placed across twenty facilities in October 2020 (17 admin interns in control facilities; 14 admin interns and 19 program interns in intervention facilities) (Fig 1). Thirty-four interns remained at 18 facilities as of August 2021 (12 in control facilities, 22 in intervention facilities). Among intervention facilities, six facilities had at least one admin intern and one program intern placed simultaneously through August 2021 (high intervention); four facilities had one admin and one program intern placed simultaneously less than half of the time (low intervention). Amongst control facilities, five had a minimum of one admin intern (and no program interns) for at least three-quarters of the study period. The other five facilities designated as controls did not meet the *a priori* definition of a control: two facilities dropped

**Table 2. Facility characteristics at baseline (January–August 2020).**

| | Control (N = 10) | Intervention (N = 10) | Total (N = 20) |
|---|---|---|---|
| **Facility Type: n (%)** | | | |
| Clinic | 7 (70%) | 5 (50%) | 12 (60%) |
| CHC | 1 (10%) | 4 (40%) | 5 (25%) |
| Hospital | 2 (20%) | 1 (10%) | 3 (15%) |
| **Counts of HIV services: mean (sd)** | | | |
| Headcount | 10,711 (3,816) | 13,854 (5,652) | 12,282 (4,963) |
| Tested for HIV | 1,815 (1,496) | 2,965 (1,867) | 2,390 (1,748) |
| Positive for HIV | 48.2 (37.3) | 71.8 (48.8) | 60.0 (44.0) |
| Initiated in 14 days | 27.1 (37.4) | 47.2 (46.5) | 37.2 (42.4) |
| Loss to follow-up | 13.2 (14.4) | 34.1 (36.8) | 23.7 (29.2) |
| **HIV service outcomes: mean (sd)** | | | |
| Ratio Tested | 0.159 (0.075) | 0.209 (0.069) | 0.184 (0.075) |
| Ratio Positive | 0.029 (0.011) | 0.024 (0.004) | 0.026 (0.009) |
| Ratio Linked 14 days | 0.461 (0.333) | 0.590 (0.268) | 0.526 (0.302) |

Sd = Standard deviation.

out at the start of the study, two had only program interns for more than half of the study, and one had admin and program interns placed simultaneously for a month (low intervention). While intervention facilities received slightly more admin interns on than intervention facilities (an average of 1.7 vs 1.4 interns/facility, respectively), attrition was greater in control clinics, leading to an equal number of admin interns serving at both intervention and control clinics over the study period. Please see the S2 Appendix for more information on intern placement, attrition, and retention.

### 3.3. Cumulative impact

There were no significant changes for outcomes when comparing intervention facilities to control facilities (Table 3). Intervention facilities did experience a greater increase in HIV testing than control facilities (Δ11.5% versus Δ5.8%, respectively), but this difference was not statistically significant (ΔΔ5.7%, p = 0.23). There was also a net increase in treatment initiation for intervention facilities as compared to control facilities, but this difference was not statistically significant (ΔΔ10.3%, p = 0.59). There was no difference in change experienced between intervention and control groups for the proportion of people testing positive for HIV or any retention outcome. However, there were notable improvements in all retention outcomes for both intervention and control groups; improvement was most pronounced for loss to follow-up, which was reduced upwards of 75% for both the intervention and control groups. There were no differences observed for the proportion of people tested for HIV who were male, youth, or male youth (Table 4).

### 3.4. Monthly variation

For most HIV service outcomes, there were minimal differences in immediate level changes and trends between the intervention and control groups (Fig 2). We did observe an immediate increase in HIV testing following intern placement in the intervention group, but none in the control group (Fig 2A); the difference between groups was statistically significant ($\beta_6$ = 8.4%

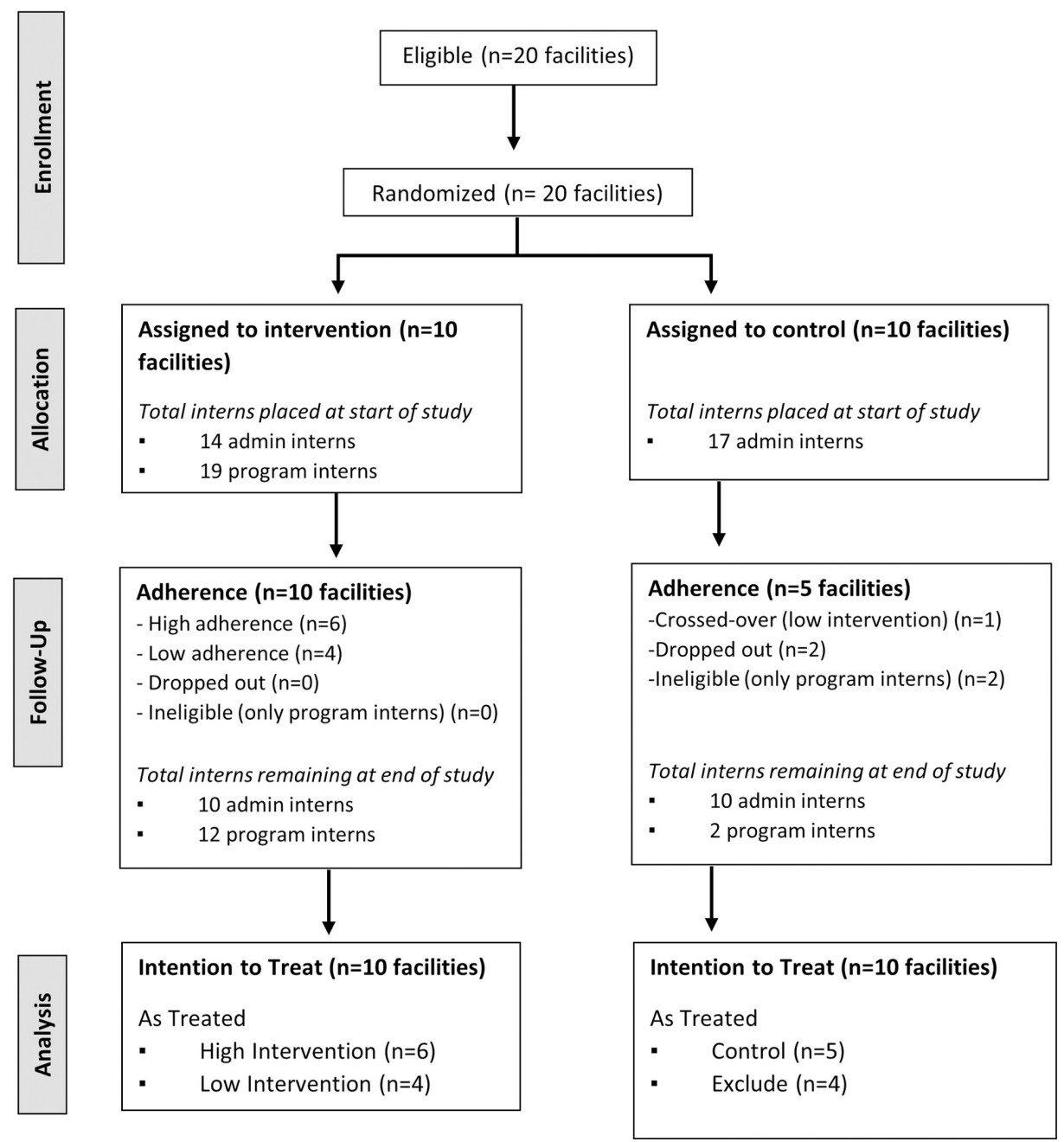

**Fig 1. CONSORT diagram for Youth Health Africa trial.**

[95% CI: 0.5–16.4%], p = 0.036). After this initial jump, the trend for HIV testing remained level for the intervention group, while it gradually increased for the control group; however, the difference in change in trend between groups was not significant ($\beta_7$ = -0.8% [-2.0%– 0.5%], p = 0.23). We also observed a steady decrease in treatment initiation for the control group after interns were placed while there was no change in trend for the intervention group, but the difference in change in trend between groups was not significant ($\beta_7$ = 0.7% [-3.2– 4.6%], p = 0.73) (Fig 2C). There were no noteworthy differences in identification of positive

**Table 3. Comparison of change in HIV service indicators between control and intervention facilities after implementation of Youth Health Africa (*Difference-in-difference analysis*).** The baseline period baseline period was January-August 2020. The study period was January-August 2021.

|  | CONTROL (n = 10) % (95% CI) | | | INTERVENTION (n = 10) % (95% CI) | | | Difference-in-Difference % (95% CI) | P-value |
|---|---|---|---|---|---|---|---|---|
|  | Baseline | Study | Difference | Baseline | Study | Difference |  |  |
| % Tested for HIV* | 15.9% (11.2–20.6%) | 21.6% (10.3–32.9%) | 5.8% (-0.9–12.4%) | 20.9% (9.6–32.2%) | 32.4% (5.1–59.7%) | 11.5% (-4.5–27.4%) | **5.7% (-3.7–15.1%)** | 0.23 |
| % Positive for HIV | 2.9% (2.3–3.4%) | 2.3% (0.9, 3.7%) | -0.5% (-1.3, 0.3%) | 2.4% (1.0–3.8%) | 1.8% (-1.6–5.2%) | -0.6% (-2.5–1.4%) | **-0.05% (-1.2–1.1%)** | 0.94 |
| % Initiated on Txt within 14 days | 46.1% (27.1–65.2%) | 35.2% (-10.8–81.3%) | -10.9% (-37.9–16.1%) | 59.0% (12.9–105%) | 58.4% (-52.8–170%) | -0.6% (-65.7–64.6%) | **10.3% (-27.8–48.5%)** | 0.59 |
| % Early Default | 10.4% (8.4–12.5%) | 10.1% (5.2–15.1%) | -0.3% (-3.2–2.6%) | 9.4% (4.5–14.4%) | 9.0% (-2.9–21.0%) | -0.4% (-7.4–6.6%) | **-0.1% (-4.2–4.0%)** | 0.94 |
| % Late Default | 5.7% (4.2–7.2%) | 4.4% (0.8–8.0%) | -1.3% (-3.4–0.8%) | 5.6% (2.0–9.2%) | 4.3% (-4.4–13.0%) | -1.3% (-6.4–3.8%) | **0.0% (-3.0–3.0%)** | 0.99 |
| % Loss to Follow-up | 9.9% (8.6–11.3%) | 2.6% (-0.7–5.8%) | -7.4% (-9.3– -5.4%) | 9.6% (6.3–12.9%) | 1.5% (-6.4–9.4%) | -8.1% (-12.7– -3.5%) | **-0.7% (-3.5–2.0%)** | 0.59 |

*Primary outcome for which the study was powered.

cases, early default, late default, or loss to follow-up between the control and intervention groups. Full model results can be found in the S3 Appendix.

## 3.5. Sensitivity analysis

The as-treated analysis yielded similar results to the intention-to-treat analysis. However, the impact on HIV testing and treatment initiation was amplified when comparing high-intervention facilities to control facilities (S4 Appendix). There was a greater difference in HIV testing between high intervention and control facilities than observed in the intention-to-treat analysis ($\beta_6$ = 13.1% [1.9–24.3%], p = 0.023), but the cumulative difference was remained insignificant ($\Delta\Delta$6.2%, p = 0.31). In addition, treatment initiation increased for high intervention facilities, resulting in a greater difference in change in trend between high intervention and control facilities than seen in the intention-to-treat analysis, but results were not significant ($\beta_7$ = 4.4% [-0.8–9.7%], p = 0.10). Differences that had been observed in the intention-to-treat analysis were generally more minor or disappeared when comparing low intervention facilities to control facilities (S5 Appendix). In the second sensitivity analysis, cumulative and monthly

**Table 4. Comparison of change in HIV testing among males and adolescents/young adults* between control and intervention facilities after implementation of Youth Health Africa.** The baseline period baseline period was January-August 2020. The study period was January-August 2021.

| Proportion tested for HIV identifying as: | CONTROL (n = 10) % (95% CI) | | | INTERVENTION (n = 10) % (95% CI) | | | Difference-in-Difference % (95% CI) | p-value |
|---|---|---|---|---|---|---|---|---|
|  | Baseline | Study | Difference | Baseline | Study | Difference |  |  |
| Male | 32.5% (28.4–36.7%) | 31.5% (21.6–41.5%) | -1.0% (-6.8–4.8%) | 31.9% (22.0–41.9%) | 30.6% (6.6–54.6%) | -1.3% (-15.4–12.7%) | **-0.3% (-8.6–7.9%)** | 0.94 |
| Adolescents or YA | 47.4% (42.6–52.1%) | 47.2% (35.8–58.7%) | -0.1% (-6.8–6.6%) | 48.9% (37.4–60.3%) | 49.3% (21.7–76.9%) | 0.4% (-15.8–16.6%) | **0.5% (-8.9–10.0%)** | 0.91 |
| Male Adolescents or YA** | 12.0% (10.3–13.8%) | 11.5% (7.3–15.8%) | -0.5% (-3.0–2.0%) | 12.2% (8.0–16.5%) | 12.0% (1.7–22.2%) | -0.2% (-6.2–5.8%) | **0.3% (-3.3–3.8%)** | 0.89 |

* Adolescents and young adults included ages 10–29 years old.

**YA: Young Adults.

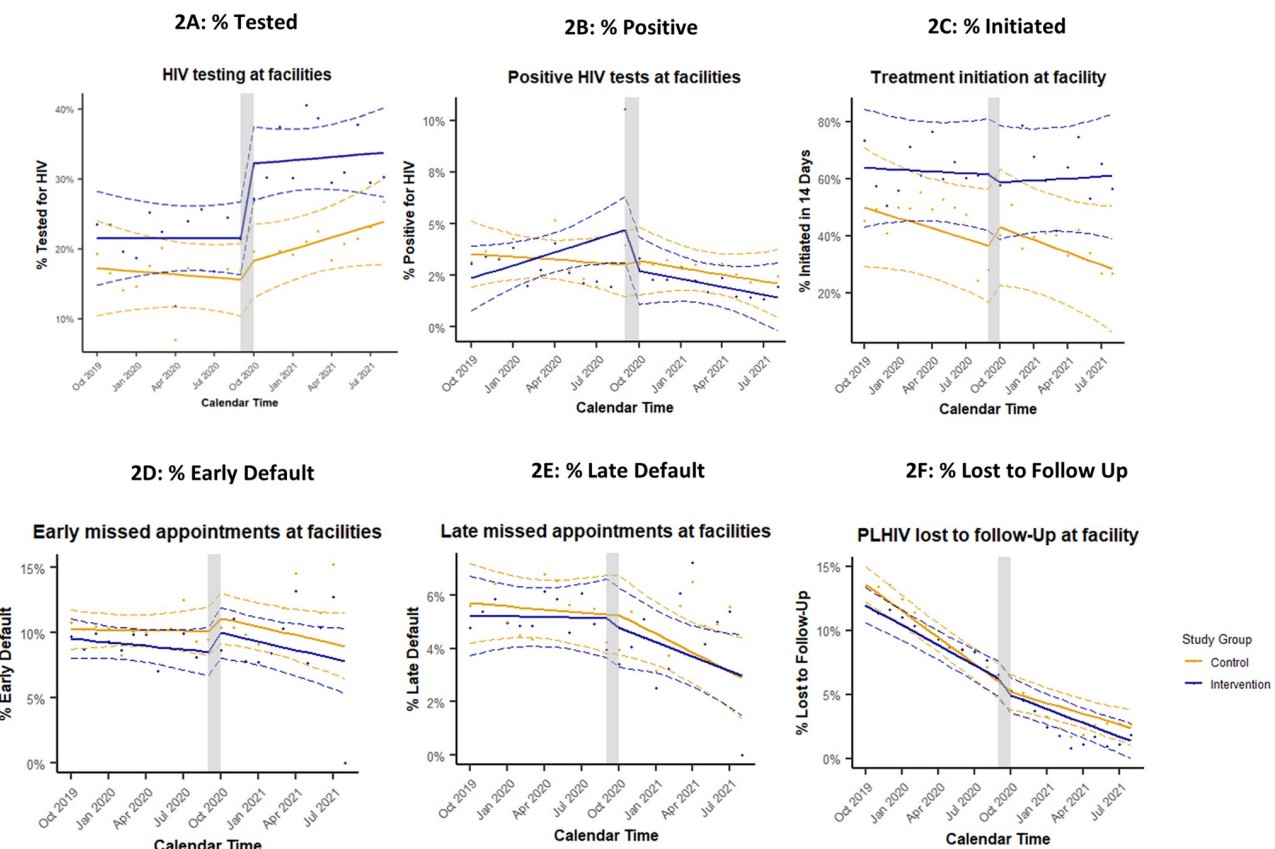

**Fig 2. Monthly reported outcomes from intervention and control facilities, interrupted by intern placement in facilities in October 2020.** Points are average outcomes per month. Solid lines represent the linear model (yellow = control, blue = intervention). Dotted lines represent the 95% confidence intervals. The grey bar indicates the start of the intervention period.

changes in proportion of people tested for HIV remained very similar to those calculated in the original analysis (S6 Appendix).

## 4. Discussion

The addition of program interns to health facilities was found to have little impact on facility-based HIV service delivery in this cluster randomized trial, although past research has suggested the program leads to improvements in HIV service delivery [12,13]. While this was largely a null trial, we did observe a significant increase in HIV testing immediately following placement of program interns and there were strong signals for positive impact on HIV testing and treatment initiation that were further amplified in the as-treated analysis. Notably, we observed no differences between control and intervention facilities in identifying HIV positive cases or retention in care. The addition of program interns to facilities also made no measurable difference in HIV testing rates for males, youth, or male youth. Evidence to reject the null hypothesis may have been insufficient due to the small sample size and limited power of this study.

While our findings may appear to contradict previous research of the YHA program, where an interrupted time series (ITS) analysis found the YHA program to be associated with significant increases in HIV testing and treatment initiation and significant reductions in loss to

follow-up [12], we hypothesize that these differences may have existed for four reasons that stem from differences in the two studies (S7 Appendix). Firstly, the randomized trial was powered to assess change in HIV testing, but it was powered to detect a larger difference than we observed; it was not powered to assess differences in other outcomes. As a result, the positive signals we found for HIV testing and treatment initiation may have been significant if our study had more power. Meanwhile, an ITS analysis is not 'powered' in the same way as a cluster randomized trial, but it included many more facilities than the trial that would make it easier to detect significant differences in outcomes.

Secondly, we hypothesize that the YHA program may be impactful, but in circumstances that were different than how it was implemented in the trial. The ITS included twenty-times more facilities than the intervention arm of the trial from numerous districts in which the intervention was implemented in a wide variety of manners (e.g., number and types of interns). The ITS analysis did not find change after the intervention in all facilities; generally, it found the largest difference in outcomes to occur among facilities with five or more interns and no differences in facilities with 1 or 2 interns. We hypothesize that the conditions of the trial—the location and size of facilities and the number of facilities placed—may not have been as conducive for change as the wider variety of conditions that existed in the ITS. For example, the intern dose used in the trial, where half of intervention facilities ended with only two interns, may have been insufficient to influence measurable change.

Thirdly, some of the differences that were observed in the ITS analysis could have been due to differences in data quality, not program change. The trial measured the impact of program interns (in addition to one or more administrative interns), whereas the ITS measured the impact of the full YHA program (program and admin interns). It is possible that we saw less impact than the ITS because it observed changes due to improvements in data quality and not just service delivery, especially for retention in care indicators [12,20]. However, healthcare workers working with YHA interns have suggested admin interns are critical in improving facility operations that could be causally linked to changes in HIV service delivery [13]. As we used admin interns as controls in our study, we may have muted the observable impact of the YHA program, specifically for retention in care outcomes, (i.e., the admin interns in both control and intervention facilities may have contributed to reductions observed for default and loss to follow-up).

Finally, differences in results between the ITS and the trial may have been due to COVID-19. The trial was implemented during the COVID-19 pandemic, while the ITS analysis reviewed data prior to this pandemic. COVID-19 could have impacted how frequently YHA interns were working in facilities. While YHA interns are essential personnel and were exempt from lockdowns, they were required to quarantine at home for up to two weeks if exposed to COVID-19, which was not an uncommon occurrence during the trial. There was also greater attrition than normal among the YHA interns in the trial, which could have been due to the stressors of COVID-19. Moreover, COVID-19 expanded the tasks that YHA program interns played at facilities during the trial. While their primary role was to support HIV programs, we know that they were also pulled into supporting COVID-19 related activities, such as COVID-19 testing. This may have diluted the time that program interns in the trial spent on traditional HIV program activities, in comparison to how time was spent by interns in the pre-COVID era. Finally, the ebb and flow of COVID-19 may have affected patient access to care, which could have resulted in challenges for testing, treatment initiation, and retention in care amongst patients in both intervention and control clinics.

In reviewing the differences observed between the trial and the ITS, we believe the trial results likely underestimated the impact of YHA on HIV service delivery. However, the trial's largely null findings are not necessarily surprising when reviewing the broader LHW literature.

The most definitive impact of LHWs on HIV programs has been seen in HIV testing [21–24], a result we observed to a small degree in this study. LHW programs have not always been found to impact health service delivery or patient outcomes, but this does not necessarily mean they had no impact on the health facility [4,5,25–27]. For example, a cluster randomized trial in South Africa found LHWs had no impact on patient outcomes but did positively affect the process of providing care [27]. A primary goal of LHW programs is to offload work from healthcare workers [3], which may mean the impact of LHWs is not always tangible [25,28]. Even if the YHA approach does not appear to impact HIV services, it could still be effective, as HCWs have described the program to reduce their workloads, improve facility operations, and benefit patient care [13]. However, we do believe that this intervention may have impacted aspects of HIV service delivery, but study limitations may have muted this observable impact.

## 4.1. Limitations

This study was subject to many limitations. Firstly, this study had a small sample size. While a larger sample size was envisioned, additional sites were not available for randomization when the study was launched, due to challenges posed by implementing a pragmatic trial during the COVID-19 pandemic. The limited number of facilities in this study coupled with the smaller level of change observed for HIV testing than initially expected resulted in this study being underpowered for the change observed in HIV testing; moreover, this study was not powered for the other outcomes we assessed, which limits the inference we can make on impact due to this intervention. Secondly, we relied on programmatic data for this analysis. While measures are in place to ensure data quality, we were not able to verify the accuracy of data used in this study. Reliance on programmatic data also limited what outcomes we could consider in this analysis; the YHA program could have impacted service delivery in ways that were not be measured by available programmatic indicators [13]. Finally, this study faced limitations as it was implemented in the context of an ongoing program. The study was conducted in a district that had a high proportion of people knowing their HIV status and high PEPFAR investment [17,18], which could limit how much impact this program could expect to have at these health facilities; if the YHA program were evaluated in districts where facilities had greater room for improvement, we may have observed greater impact. Similarly, the pragmatic nature of this trial meant there was little control over the intervention once implemented. There were frequent changes in intern numbers and roles that were outside the control of the research team. Differences observed in the as-treated analysis suggests the lack of time certain facilities spent with the full intervention reduced the impact observed. There was also high intern attrition, possibly because of COVID, leading to lower numbers of interns per facility than anticipated. This may have reduced impact of the intervention, as past research has found that impact is more likely observed when higher numbers of interns are present [12]. Overall, this study reveals the limitations of implementing randomized trials as part of ongoing programs and highlights the comparative strengths of using quasi-experimental designs to assess impact of programs as implemented.

## 5. Conclusion

This pragmatic cluster randomized trial found program interns from the novel YHA program to have little impact on HIV service delivery at facilities in South Africa. There were significant, immediate increases in HIV testing observed after program interns started in facilities, but there was insufficient evidence to draw robust inference on the program's impact on improving indicators related to HIV positive cases, treatment initiation, retention in care, or testing among males or youth. While this was largely a null trial, we do not believe this suggests the

YHA program had no impact health facilities. Rather, when we review these results alongside previous research [12,13], we believe study limitations, specifically intervention adherence and limited study power, could have muted or negated impact we observed and that this program should still be considered for roll-out. In particular, YHA may impact HIV service delivery if there are a high enough number of interns placed at the facility, which we may not have achieved in this study. Moving forward, there is a need to further elucidate the circumstances under which YHA provides its greatest impact and to reassess the indicators that should be used to measure the program's impact on the health facility.

## Supporting information

**S1 Appendix. CONSORT checklist.**
(PDF)

**S2 Appendix. Trial monitoring data.**
(PDF)

**S3 Appendix. Model parameters for time series analyses (Intention-to-Treat).**
(PDF)

**S4 Appendix. Sensitivity Analysis—As-Treated Results (High Intervention vs Control facilities).**
(PDF)

**S5 Appendix. Sensitivity Analysis—As-Treated Results (Low Intervention vs Control facilities).**
(PDF)

**S6 Appendix. Sensitivity Analysis—% Tested.**
(PDF)

**S7 Appendix. Comparison to previously published research.**
(PDF)

**S1 File.**
(ZIP)

**S2 File.**
(DOCX)

**S3 File.**
(PDF)

## Acknowledgments

We thank the NMM District Health Officer for his roles in supporting facility identification and randomization for this study and the staff at Youth Health Africa for their role in implementing this program and tracking intern roles for the study period.

## Author Contributions

**Conceptualization:** Deanna Tollefson, Sayan Dasgupta, Geoffrey Setswe, Sarah Reeves, Salome Charalambous, Ann Duerr.

**Data curation:** Deanna Tollefson, Sarah Reeves.

**Formal analysis:** Deanna Tollefson.

**Methodology:** Deanna Tollefson, Sayan Dasgupta, Salome Charalambous, Ann Duerr.

**Project administration:** Geoffrey Setswe, Sarah Reeves.

**Supervision:** Geoffrey Setswe, Salome Charalambous, Ann Duerr.

**Validation:** Sayan Dasgupta.

**Visualization:** Deanna Tollefson.

**Writing – original draft:** Deanna Tollefson.

**Writing – review & editing:** Deanna Tollefson, Sayan Dasgupta, Geoffrey Setswe, Sarah Reeves, Salome Charalambous, Ann Duerr.

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
