## [Decision Letter · Decision Letter 0]

1 Aug 2023

PONE-D-23-12962Impact of youth lay health workers on HIV service delivery in South Africa: a pragmatic cluster randomized trialPLOS ONE

Dear Dr. *Deanna Tollefson*,

Thank you for submitting your manuscript to PLOS ONE. After careful consideration, we feel that it has merit but does not fully meet PLOS ONE’s publication criteria as it currently stands. Therefore, we invite you to submit a revised version of the manuscript that addresses the points raised during the review process.

We look forward to receiving your revised manuscript.

Kind regards,

Prof. Samuel Bosomprah, PhD

Academic Editor

PLOS ONE

Journal Requirements:

3. We note that you have selected “Clinical Trial” as your article type. PLOS ONE requires that all clinical trials are registered in an appropriate registry (the WHO list of approved registries is at https://www.who.int/clinical-trials-registry-platform/network/primary-registries"" https://www.who.int/clinical-trials-registry-platform/network/primary-registries" https://www.who.int/clinical-trials-registry-platform/network/primary-registries and more information on trial registration is at http://www.icmje.org/about-icmje/faqs/clinical-trials-registration/). Please state the name of the registry and the registration number (e.g. ISRCTN or ClinicalTrials.gov) in the submission data and on the title page of your manuscript. a) Please provide the complete date range for participant recruitment and follow-up in the methods section of your manuscript. b) If you have not yet registered your trial in an appropriate registry, we now require you to do so and will need confirmation of the trial registry number before we can pass your paper to the next stage of review. Please include in the Methods section of your paper your reasons for not registering this study before enrolment of participants started. Please confirm that all related trials are registered by stating: “The authors confirm that all ongoing and related trials for this drug/intervention are registered”. Please see http://journals.plos.org/plosone/s/submission-guidelines#loc-clinical-trials for our policies on clinical trials.

"The Aurum Institute was a funder to the Youth Health Africa organization, and author SR supported monitoring and evaluation activities for Youth Health Africa. No other authors have any conflicts of interest to report."

Reviewers' comments:

Reviewer's Responses to Questions

**Comments to the Author**

1. Is the manuscript technically sound, and do the data support the conclusions?

Reviewer #1: Partly

Reviewer #2: Partly

2. Has the statistical analysis been performed appropriately and rigorously? 

Reviewer #1: Yes

Reviewer #2: Yes

3. Have the authors made all data underlying the findings in their manuscript fully available?

Reviewer #1: Yes

Reviewer #2: Yes

4. Is the manuscript presented in an intelligible fashion and written in standard English?

Reviewer #1: Yes

Reviewer #2: Yes

5. Review Comments to the Author

Reviewer #1: This is an interesting pragmatic, cluster randomized trial among 20 facilities in Ngaka Modiri Molema district in Northwest province of South Africa over the period October 2020-August 2021 to assess impact of YHA interns on HIV testing, treatment initiation, and retention in care. The paper is generally well written and the Discussion articulated.

However, there are some points than need to be further addressed.

Major points

1. Was HIV-RNA available to measure? Unclear why this outcome was not considered for the treated in Table 1 besides retention in care.

2. As briefly discussed by the authors, power of the analysis is a concern. Researchers seems to have overestimated the underlying rate of HIV testing in these facilities. Indeed, they were expecting a rate of 18% while only 7% was the observed. This impacted on statistical power, regardless of the number of facilities involved.

3. There is nothing particularly wrong with the analysis although the methodology used (difference in difference) is typical of observational studies in which there is no exchangeability at baseline. Although, some differences between intervention and controls facilities have been observed, none appeared particularly large, so it is unclear why a simpler analysis comparing the outcomes at the end of the study period comparing intervention and controls was not conducted.

4. I would avoid reporting p-values in Table 2. In addition, some of the reported figures are proportions. T-test is not a valid test to compare proportions.

5. I would have liked to see if control and intervention facilities were balanced with regards of admin interns at baseline (in terms of number interns/total head count at the facility) and a more formal comparison of attrition of interns over time (if data available).

6. The key part of Tables 3,4 (DID analysis) is cut out from the pdf document. This is key for a correct interpretation as it shows the uncertainty around the DID estimates (not reported in the text).

7. Unclear whether the decision to look at the subsets of males adolescents was taken a priori or not. In any case it would have been advisable to formally test for interaction before showing the stratified results.

8. First paragraph of the Discussion/ final Conclusions. Because of the limited power I would be cautious in concluding that the intervention ‘had little impact’ or that it was a ‘null’ trial. Differences in some outcomes were quite large so it is possible that the trial was inconclusive due to limited power. No evidence for an effect does not mean no effect. The interpretation of the subset analysis is even more tricky as the study was not adequately powered to detect these in first place.

Minor points

1. I would avoid words like ‘significant’, ‘insignificant’ etc. and rephrase in the context of evidence against the null hypothesis.

2. Although the impact of quarantine/lockdown due to COVID-19 is a fair point, the sensitivity analysis restricted to facilities in which the interns were present throughout most of the study period should have mitigated this bias. The interference of COVID-19 care seems to be a more valid point.

3. Besides the fact that admin interns might have impacted on the rate of retention in care in both arms hence diluting the difference which could be other explanation for the larger difference between arms seen in rate of HIV testing alone? And why the waning over time?

4. Some of the arguments in the main Discussion are repeated in the Limitation section (power, impact of COVID-19 etc). I would avoid these repetitions to improve reading flow.

Reviewer #2: The study quantified the impact of youth lay health workers on HIV service delivery in South Africa using a pragmatic cluster randomized trial.

These are my comments

1) Is the intervention Youth Health Africa (YHA) program or youth lay health workers on HIV. Is a bit confusing because in the abstract you indicated that you are assessing the impact of YHA interns on HIV testing, treatment initiation, and retention in care. The name of the intervention must be reflected in the title.

2) Can you include the 95% confidence interval for the impact on HIV testing in the abstract. You only indicated 10.1% which is just the point estimate and doesn’t say much about the true effect size.

3) Major comment: On what basis did you arrive at 20 health facilities (10 facilities in each arm)? Is your power analysis based on the number of health facilities (control versus intervention) or based on the number of individuals assigned to these health facilities. Since the intervention is creating clusters, your power analysis should be based on the number of health facilities not the number of individuals within the health facilities. We prefer more clusters (facilities randomized into intervention and controls) compared to the number of individuals within the facilities due to correlation. Within each facility, how many individuals (interns) were placed and what was the justification? This section is very scanty, and detail must be provided.

4) You admitted that the 20 facilities were purposively selected which clearly means that your study is not externally valid because the 20 facilities will not be representative of all target health facilities. Can the study be recommended for a larger scale up?

5) You indicated that the study had 80% power to detect a minimum of 35% change for HIV testing if the intraclass correlation was 0.01 (i.e., change from 18.4% people tested to 24.5% of people tested), but a change of 18.4% people tested to 24.5% is like 6.1 percentage point increase, so why 35% change for HIV testing. Please explain.

Results and discussions are okay on condition the point raised in the method section are addressed. They follow the CONSORT checklist.

6. PLOS authors have the option to publish the peer review history of their article (what does this mean?). If published, this will include your full peer review and any attached files.

Reviewer #1: No

Reviewer #2: No

---

## [Author Response · Author response to Decision Letter 0]

9 Oct 2023

*Please see attached cover letter and the response to reviewers file*

---

## [Decision Letter · Decision Letter 1]

7 Nov 2023

Impact of youth lay health workers on HIV service delivery in South Africa:  a pragmatic cluster randomized trial of the Youth Health Africa program

PONE-D-23-12962R1

Dear Dr. *Tollefson*,

We’re pleased to inform you that your manuscript has been judged scientifically suitable for publication and will be formally accepted for publication once it meets all outstanding technical requirements.

Kind regards,

Prof. Samuel Bosomprah, PhD

Academic Editor

PLOS ONE

Additional Editor Comments (optional):

Reviewers' comments:

Reviewer's Responses to Questions

**Comments to the Author**

1. If the authors have adequately addressed your comments raised in a previous round of review and you feel that this manuscript is now acceptable for publication, you may indicate that here to bypass the “Comments to the Author” section, enter your conflict of interest statement in the “Confidential to Editor” section, and submit your "Accept" recommendation.

Reviewer #2: All comments have been addressed

2. Is the manuscript technically sound, and do the data support the conclusions?

Reviewer #2: Yes

3. Has the statistical analysis been performed appropriately and rigorously? 

Reviewer #2: Yes

4. Have the authors made all data underlying the findings in their manuscript fully available?

Reviewer #2: Yes

5. Is the manuscript presented in an intelligible fashion and written in standard English?

Reviewer #2: Yes

6. Review Comments to the Author

Reviewer #2: Authors have adequately addressed all comments especially queries on the sampling design and other technical issues.

7. PLOS authors have the option to publish the peer review history of their article (what does this mean?). If published, this will include your full peer review and any attached files.

Reviewer #2: No

---

## [Editor Report · Acceptance letter]

22 Nov 2023

PONE-D-23-12962R1 

Impact of youth lay health workers on HIV service delivery in South Africa:  a pragmatic cluster randomized trial of the Youth Health Africa program 

Dear Dr. Tollefson:

I'm pleased to inform you that your manuscript has been deemed suitable for publication in PLOS ONE. Congratulations! Your manuscript is now with our production department. 

Kind regards, 

on behalf of

Prof. Samuel Bosomprah 

Academic Editor

PLOS ONE